# Immune System and Brain/Intestinal Barrier Functions in Psychiatric Diseases: Is Sphingosine-1-Phosphate at the Helm?

**DOI:** 10.3390/ijms241612634

**Published:** 2023-08-10

**Authors:** David Martín-Hernández, Marina Muñoz-López, Hiram Tendilla-Beltrán, Javier R. Caso, Borja García-Bueno, Luis Menchén, Juan C. Leza

**Affiliations:** 1Departamento de Farmacología y Toxicología, Facultad de Medicina, Universidad Complutense de Madrid (UCM), Instituto de Investigación Hospital 12 de Octubre (i+12), Instituto Universitario de Investigación en Neuroquímica (IUIN), 28040 Madrid, Spain; marmun29@ucm.es (M.M.-L.); jrcaso@med.ucm.es (J.R.C.); bgbueno@med.ucm.es (B.G.-B.); jcleza@med.ucm.es (J.C.L.); 2Centro de Investigación Biomédica en Red de Salud Mental, Instituto de Salud Carlos III (CIBERSAM, ISCIII), 28029 Madrid, Spain; 3Laboratorio de Neuropsiquiatría, Instituto de Fisiología, Benemérita Universidad Autónoma de Puebla (BUAP), 72570 Puebla, Mexico; hiramtb20@gmail.com; 4Servicio de Aparato Digestivo, Hospital General Universitario Gregorio Marañón, Departamento de Medicina, Universidad Complutense, Instituto de Investigación Sanitaria Gregorio Marañón, 28007 Madrid, Spain; luisalberto.menchen@salud.madrid.org; 5Centro de Investigación Biomédica en Red de Enfermedades Hepáticas y Digestivas, Instituto de Salud Carlos III (CIBEREHD, ISCIII), 28029 Madrid, Spain

**Keywords:** sphingosine-1-phosphate, psychiatric diseases, immune system, inflammation, barrier function, blood–brain barrier, intestinal barrier

## Abstract

Over the past few decades, extensive research has shed light on immune alterations and the significance of dysfunctional biological barriers in psychiatric disorders. The leaky gut phenomenon, intimately linked to the integrity of both brain and intestinal barriers, may play a crucial role in the origin of peripheral and central inflammation in these pathologies. Sphingosine-1-phosphate (S1P) is a bioactive lipid that regulates both the immune response and the permeability of biological barriers. Notably, S1P-based drugs, such as fingolimod and ozanimod, have received approval for treating multiple sclerosis, an autoimmune disease of the central nervous system (CNS), and ulcerative colitis, an inflammatory condition of the colon, respectively. Although the precise mechanisms of action are still under investigation, the effectiveness of S1P-based drugs in treating these pathologies sparks a debate on extending their use in psychiatry. This comprehensive review aims to delve into the molecular mechanisms through which S1P modulates the immune system and brain/intestinal barrier functions. Furthermore, it will specifically focus on psychiatric diseases, with the primary objective of uncovering the potential of innovative therapies based on S1P signaling.

## 1. Immune Dysregulation and Barrier Function in Psychiatric Diseases

### 1.1. Immune Dysregulation in Psychiatric Diseases

A substantial proportion of psychiatric patients (20–60%) fail to achieve a complete remission with currently available treatments, a statistic that remains consistent for severe mental illnesses (SMI), including major depressive disorder (MDD) and schizophrenia (SZ) [1]. This situation has prompted the scientific community to explore new pathways in the pathophysiology of these SMI, which may unveil new targets, diagnostic tools, or risk biomarkers to personalize and stratify treatments. In this context, a promising area of research in recent years has revolved around the potential dysregulation of immune or inflammatory responses in both the brain and the rest of the body in these diseases. Evidence from both preclinical models and studies involving human samples suggests a pathophysiological role of these processes. Importantly, therapeutic interventions could potentially target these pathways to enhance patient treatment outcomes.

Neuroinflammation is a common feature of many neurodegenerative diseases, and over the last two decades, this concept has also extended to psychiatric disorders. To illustrate this, we will go into details with MDD and SZ, but immune alterations have been also found in other mental pathologies, including such as bipolar disorder (BD) [2], autism spectrum disorder (ASD) [3], attention deficit hyperactivity disorder (ADHD) [4], and eating disorders (EDs) [5].

The first genome-wide association studies (GWAS) [6] identified a substantial number of genes (108) implicated in cohorts of approximately 40,000 patients with SZ. Several other genetic association studies have validated these data. Among these genes, a prominent cluster (the largest, surpassing other neurotransmitter-related ones) located on chromosome 6 may regulate the inflammatory and immune response. This discovery has confirmed previous clinical evidence and underscored the connection between the immune system and SZ, a relationship further corroborated in MDD [7,8].

Among the nonspecific inflammatory features identified in blood analyses of psychiatric patients, leukocytosis with neutrophilia is one of the most consistent in first-episode psychosis, SZ, and BD [9,10]. Another nonspecific inflammatory parameter elevated in SZ patients is the erythrocyte sedimentation rate [11]. Interestingly, longitudinal studies encompassing tens of thousands of individuals tracked over a period of 45 years (who had their erythrocyte sedimentation rate measured during health examinations before military service), revealed that those who later developed psychotic symptoms or SZ exhibited higher levels of this nonspecific inflammatory parameter years earlier. This suggests a potential sensitizing effect of prior low-grade inflammatory stimulus in vulnerable individuals.

Hence, innate and adaptive immune alterations have been observed in MDD [12,13,14] and SZ [15,16,17]. Moreover, some studies support the existence of psychiatric and autoimmune comorbidities [18]. Imaging studies (in vivo, positron emission tomography, or postmortem tissue evaluation for anatomopathological assessment) have revealed microglial activation in the brains of patients with MDD [19,20] or SZ [21,22]. Interestingly, there is a correlation between systemic inflammation and brain functional consequences: patients with higher plasma C-reactive protein (CRP) levels showed fewer connectivity indices in the central nervous system (CNS) [23].

It is well-established that certain immunomodulatory treatments, such as interferon therapy, may induce MDD symptoms [24]. Recent evidence also indicates the development of SZ-related behaviors under similar circumstances [25]. Similarly, the administration of cytokines or an immunological challenge has caused behavioral changes. For instance, in humans, injection of lipopolysaccharide (LPS), the main component of the outer membrane of Gram-negative bacteria, induces sickness behavior (depressed mood, anhedonia, anxiety, etc.), systemic cytokine increases, and microglial activation [26]. Behavioral alterations resembling MDD or SZ, triggered by various immune stimuli—such as LPS and polyinosinic: polycytidylic acid (Poly I:C)—have served to develop preclinical models for the study of these pathologies [27].

Given the existing evidence, there exists a bidirectional relationship between the immune system and current pharmacological treatments: antidepressants and antipsychotics have demonstrated anti-inflammatory effects beyond their conventional neurotransmitter actions [28,29], while anti-inflammatory drugs have shown antidepressant and antipsychotic effects or the ability to enhance the efficacy of these drugs when used as adjuvants [30,31,32]. Nevertheless, the question of whether low-grade inflammation or immune dysregulation is a cause or consequence of mental health disorders remains an area of ongoing debate.

The study by Kappelmann et al. [11], mentioned earlier, suggested a potential causal link in individuals entering military service. Moreover, a long-term study in the Avon County of the United Kingdom has reported an “imprinting” effect [33]. Children exhibiting higher levels of IL-6 at the age of 9 were at increased risk of developing psychosis or psychotic symptoms by the age of 18. In another study conducted in Denmark, involving nearly 80,000 individuals followed over several years, patients with SZ or psychotic symptoms displayed elevated inflammatory parameters, such as C-reactive protein (CRP). Specifically, individuals in the fourth quartile of CRP levels had the highest likelihood of developing the disease. These findings hold considerable statistical robustness, as the difference persists in sub-analyses adjusted for sex and age, and also in multivariate analyses [34]. However, the relationships with depressive disorders are not as clearly defined.

The possibility of a widespread inflammatory environment throughout the system reaching the brain via a more permeable blood–brain barrier (BBB) remains an intriguing open question in numerous ongoing research projects worldwide. The origin of these inflammatory elements in circulation is still a topic of debate, whether they stem from a mobilization from cellular nests or from the activation of immune cells by stimulating molecules. In this regard, recent preliminary data indicate that patients with SZ or MDD showed an increase in the permeability of the intestinal barrier [35,36] and vice versa, patients with inflammatory bowel diseases were susceptible to psychiatric symptoms through leaky gut, a phenomenon related to increased intestinal permeability [37]. Moreover, recent GWAS revealed a shared genetic background between psychiatric and gastrointestinal disorders [38]. This concurrence of phenomena may signify the potential translocation of bacteria-derived LPS from the colon, as demonstrated in preclinical settings using stress protocols [39].

Studies by Cai et al. [40] have recently shown that LPS can reach the BBB in mental disorders. Specifically, there is an increase in perivascular BBB macrophages in postmortem specimens, along with elevated inflammation markers, such as vascular cell adhesion molecule (VCAM), identified in perivascular areas and astrocytes. Consequently, both the brain and intestinal barriers emerge as crucial structures in psychiatric diseases, and their dysfunctions may play a pivotal role in driving their pathophysiology.

### 1.2. The Brain Barriers in Psychiatric Diseases

#### 1.2.1. Structure and Function of the Brain Barrier

The conventional notion of a singular BBB has evolved into the recognition of several distinct brain barriers, each occupying specific neuroanatomic locations and comprising unique structural components that regulate interactions between brain parenchyma and blood-born molecules or cells [41,42]:BBB: constituted by microvascular endothelial cells with apical tight junction complexes lining the cerebral capillaries that traverse the brain and spinal cord. Heterogenic endothelial cells shape the neurovascular complex interacting with neurons, astrocytic endfeet, myocytes, perivascular macrophages, pericytes, and extracellular matrix components [43].Arachnoid (or meningeal) barrier: formed by the avascular arachnoid epithelium, featuring tight junctions as a barrier between the cerebrospinal fluid (CSF)-filled subarachnoid space and other meningeal structures.Blood-CSF barrier: this barrier, also constituted by epithelial cells with apical tight junctions, separate the blood vessels of the choroid plexus (fenestrated and leaky) from the CSF.The fetal CSF-brain barrier: during early development, this barrier exists between the CSF and brain parenchyma during early development. It acts as a functional barrier when junctions interconnect ependymal cells.The adult ventricular ependyma: as development progresses, the ependymal cells lose their ability to restrict the passage of larger molecules, such as proteins, between the CSF and the brain.

There are additional brain-blood interfaces known as the circumventricular organs (CVOs), which are characterized by the absence of the BBB and are highly vascularized with fenestrations [43]. These brain structures, located in the third and fourth ventricles, can classify as sensory (subfornical organ, organum vasculosum of the lamina terminalis, and area postrema) or secretory (subcommissural organ, pituitary neural lobe, median eminence, and pineal gland) CVOs [43].

Different types of junctions participate in connecting endothelial cells of the BBB. Intercellular tight junctions (TJs) provide firm mechanical stability with low transcytosis rates and composed of proteins spanning the intercellular gap (occludins and claudins), junctional adhesion molecules (JAMs), and regulatory proteins (Zonula Occludens 1, 2, 3 (ZO-1, ZO-2, ZO-3)), responsible for linking transmembrane proteins with the actin cytoskeleton [44]. Adherens junctions consist of cadherins and catenins that link the actin cytoskeleton with cadherins, while gap junctions encompass tissue-specific isomers of the connexin (Cx) family. Collectively, these junctional proteins ensure strict regulation of paracellular permeability in the BBB [45].

Transporters are another vital component of the BBB structure, regulating the transport of nutrients, neuroactive peptides, large proteins, and other molecules into and out of the brain through multiple mechanisms, including carrier-mediated transport (facilitated diffusion), receptor-mediated transport systems, and active efflux transporters (ATP-binding cassette (ABC) transporter superfamily [46]. Transcytosis, another transport mechanism across the brain endothelium, relies on vesicle trafficking and is strictly controlled under physiological conditions. Three types of transcytosis regulate the transcellular transport of lipophobic molecules across the cerebral endothelium: fluid phase, absorptive, and receptor-mediated (clathrin and caveolae dependent) [47]. Lastly, another important transport mechanism associated with the BBB is the glymphatic system, which primarily facilitates debris clearance by controlling the exchange of CSF and interstitial fluid through astrocytic cells [48].

#### 1.2.2. Brain Barrier Dysfunction in Psychiatric Diseases

One of the hallmarks of neuroinflammation is the dysregulation in the structure and function of brain barriers, leading to a loss of barrier integrity and increased general permeability to molecules and cells. Dysfunction of the BBB is a crucial event in the pathophysiology of neurological conditions such as traumatic brain injury or stroke [49,50]. However, the investigation of potential structural and functional alterations in the BBB in neuropsychiatric diseases such as SZ, MDD, or ASD is still ongoing, probably due to varying degrees of inflammatory processes among the affected patient clusters (but not all) [51,52].

Studies have predominantly focused on alterations in the structure and function of proteins forming endothelial TJs, such as claudin-5, in neuropsychiatric disorders [53,54]. Additionally, other changes have been identified in preclinical and clinical studies using biomarkers, postmortem tissue, and neuroimaging [55]. Preclinical data are massive, and this review summarizes the principal clinical evidence:Increased paracellular traffic of macromolecules (albumin, IgG) [56]Upregulation of cell adhesion molecules (integrins, cadherins) [57,58] facilitating paracellular and transcellular pathways for the CNS infiltration of peripheral leukocytes (T and B lymphocytes) [59]Elevated serum and CSF levels of astrocyte-derived neurotrophin S100B, serving as a marker for brain damage, BBB disruption, and even brain injury (alarmin) [60,61]Structural and functional changes of cellular populations located in the perivascular space (pericytes, border-associated macrophages) [40,62,63,64]Dominant microglial activation in brain parenchyma, creating an inflammatory context that may exacerbate BBB permeability [65,66]Changes in the expression and function of P-glycoprotein [67] and other transcellular transporters (transferrin receptor (TfR), glucose transporter-1 (GLUT-1), insulin receptor (IR), and low-density lipoprotein receptor (LDLr)), which may impact the access of neuropsychiatric drugs and other promising pharmacological tools (e.g., nanoparticles, viral vectors) to the brain parenchyma through the BBB, as proposed for the treatment of neurological/neurodegenerative diseases [68,69].

Furthermore, the breakdown of the glymphatic system, mainly described in neurodegenerative diseases, is currently under investigation in psychiatric research due to its relevance to neuroinflammation and BBB permeability [70].

### 1.3. The Intestinal Barrier in Psychiatric Diseases

#### 1.3.1. The Intestinal Barrier in Health and Disease

Besides its digestive and absorptive roles, the gastrointestinal tract exerts an effective but dynamic barrier function between the mucosal immune system and the vast array of microbial and alimentary antigens present in the intestinal lumen. Central to this barrier is the intestinal epithelial cells (IECs), which play vital roles both in generating immune tolerance and orchestrating effective innate and adaptive immune responses. The intestinal barrier encompasses not only IECs but also a variety of non-cellular elements including mucin, antimicrobial peptides, secretory immunoglobulin A (sIgA), and intercellular junction molecules between adjacent IECs [71,72].

Inflammatory cytokines, such as interferon gamma (IFN-γ) and tumor necrosis factor alpha (TNF-α), can influence the anatomy and function of intercellular junctions [73], with a subsequent increased barrier permeability. This inflammation-associated dysfunction of the intestinal barrier can result in the translocation of commensal microorganisms and microbial products to the lamina propria, lymphatic vessels, and portal venous system. This translocation may trigger and perpetuate local and even systemic inflammatory processes. As a matter of fact, increased intestinal permeability is an early pathophysiological event in humans [74] and experimental [75] inflammatory bowel disease (IBD).

Additionally, experimental models of physical and psychological stress have demonstrated disruption of intestinal tight junctions, assuming their direct responsibility on the increased intestinal permeability observed in stressed laboratory animals [76]. In this sense, stressful events could trigger or modify the clinical course of IBD [77]. Moreover, the prevalence of psychiatric symptoms in IBD patients is substantial, with anxiety estimated to affect one-third and depression to one-quarter, especially those with active disease [78]

#### 1.3.2. Intestinal Barrier Dysfunction in Psychiatric Diseases

Despite considerable efforts dedicated to studying the relationship between inflammatory balance and psychiatric pathology, there remain significant gaps in our understanding. Firstly, genetic vulnerability itself is insufficient for the onset of these diseases, suggesting the relevance of epigenetic changes induced by environmental factors, including inflammation. Secondly, the inflammatory response is nonspecific; several stimuli can trigger similar alterations in different psychiatric diseases, with slight nuances depending on whether it occurs in an early or chronic stage [79]. Thirdly, extensive longitudinal studies have failed to elucidate whether the activation of the immune system and the inflammatory response are the cause or consequence of these mental illnesses. Lastly, adjuvant anti-inflammatory therapies lead to moderate and non-stable symptom improvement, as inflammation is a homeostatic mechanism whose blockade could have counterproductive effects due to shared intra and intercellular pathways between pro- and anti-inflammatory mechanisms [80,81].

The origin of neuroimmune alterations in psychiatric diseases remains elusive. Epidemiological evidence suggests that peripheral immune alterations caused by circulating endotoxins through leaky internal barriers, including the intestinal barrier, could initiate an immune/inflammatory cascade in the brain, leading to structural or functional brain damage. This low-grade inflammation status, not an evident infection, seems a common factor in chronic diseases such as MDD and SZ [82]. At least in the initial years of the disease, stress is a prominent driving factor associated with inflammation in these pathologies. Other contributing factors include smoking, poor hygiene, diet, and metabolic changes induced by the disease or medications. However, another immune system mediator might play an important role in driving this inflammatory phenotype in psychiatric disorders: endotoxins.

Leaky gut and subsequent bacterial translocation have been identified clinically in MDD and SZ [83,84]. Recent research described different bacterial genera in fecal samples from MDD patients [85]. In SZ, leaky gut was significantly associated with a negative phenotype [86]. Stress-based experimental models of MD and SZ reported increased colony-forming units (CFUs) in tissues adjacent to the colon and elevated levels of LPS and LBP in plasma [87]. Structural and functional studies demonstrated that stress induces disruption of the intestinal barrier [39,88], and LPS from *E Coli* was observed to interact with innate immune receptors, such as Toll-like receptors (TLRs) expressed in brain areas [89]. Pharmacological and genetic studies using knockout mice indicate a role for endotoxins in depressive-like behaviors, identifying Gram-negative, Gram-positive, and anaerobic genera associated with bacterial translocation [39,87].

Several mechanisms have been proposed for the contribution of bacterial translocation to neuroimmune alterations in the CNS [90]. Peripheral proinflammatory cytokines associated with leaky gut may communicate with the brain via (a) the neural pathway, involving systemic cytokines directly activating primary afferent nerves (mainly the vagus); (b) the humoral pathway, through the choroid plexus and circumventricular organs (CVOs), which lack an intact BBB, allowing circulating cytokines to enter into the cerebral parenchyma through volume diffusion and elicit downstream signaling events important in altering brain function; (c) the cellular pathway, where systemic proinflammatory cytokines activate endothelial cells, leading to the activation of adjacent perivascular macrophages, which in turn activate microglia [91]. Additionally, LPS or other bacteria components from the gut can gain access to the circulation and directly invade the brain through a more permeable BBB or signal through leptomeningeal cells. Those cells express TLRs for LPS, and the released proinflammatory cytokines activate microglia, evoking neuroinflammation.

## 2. The Sphingosine-1-Phosphate (S1P) Metabolism and Signaling

Sphingosine-1-phosphate (S1P) is a bioactive sphingolipid that plays a crucial role in the immune system. It is widely expressed throughout the body and is involved in immune activation and regulation of cellular trafficking [92]. In addition to its immunological functions, S1P also influences other essential cellular processes, including barrier integrity, angiogenesis, and proliferation, through its synthesis in platelets, erythrocytes, vascular endothelial cells, and hepatocytes [93].

Many of the S1P actions are mediated through “inside-out signalling” wherein it acts as a ligand for a group of cell surface receptors belonging to the G-protein-coupled receptors (GPCRs) superfamily. S1P is transported to the extracellular milieu by specific proteins, such as sphingolipid receptor or spinster 2 (Spns2) [94], ATP-Binding cassette (ABC) transporters A1 (ABCA1) and C1 (ABCC1) [95,96], and major facilitator superfamily transporter 2b (Mfsd2b) [97]. These mechanisms allow S1P to exert its diverse effects on cellular responses and physiological processes throughout the body.

### 2.1. S1P Metabolism

Numerous studies have evinced the significance of phospholipids and their metabolites in various pathological conditions, including cardiovascular disorders, oncogenesis, and inflammatory diseases such as multiple sclerosis (MS) and IBD [97,98,99,100]. Among lipids, sphingolipids hold particular importance, characterized by the presence of sphingosine as their common base, which can be synthesized de novo or derived from a complex lipid hydrolysis pathway [101]. Ceramide (CER) plays a pivotal role in sphingolipid metabolism as the precursor of sphingosine, catalyzed by ceramidases [99]. CER and S1P exhibit opposing functions and the establishment of a CER/S1P rheostat is crucial for maintaining S1P gradients necessary for immune cell migration processes [102]. In humans, five ceramidases have been identified: acid ceramidase, neutral ceramidase, and alkaline ceramidase 1, 2, and 3, encoded by five different genes (ASAH1, ASAH2, ACER1, ACER2, and ACER3) [103]. Lastly, sphingosine kinases (Sphks) phosphorylate sphingosine to form S1P [104].

Of note, S1P synthesis is a complex mechanism characterized by its compartmentalization in different subcellar spaces due to its hydrophobic nature, which limits its metabolism to local enzymes. Consequently, S1P can exhibit diverse signaling properties depending on its cellular location [105]. The specific localization of S1P synthesis within distinct subcellular compartments gives rise to a complex network of vesicular and active protein mechanisms that regulate the transport of S1P across these compartments. As S1P is synthesized and localized in specific organelles, such as the endoplasmic reticulum (ER) and Golgi apparatus, it undergoes specific enzymatic modifications and interactions with other molecules before being transported to its intended target sites. These modifications and interactions can further regulate the signaling properties of S1P and contribute to its diverse cellular effects. Notably, in various tissues, intracellular S1P concentrations are maintained at low levels through rapid degradation facilitated by the S1P lyase (SGPL1) enzyme present in the ER. However, in contrast to this degradation pathway, S1P can also be transported out of the cell through specific transporters, enabling it to exert effects outside the cell [106].

#### 2.1.1. S1P Synthesis Enzymes

To date, two distinct Sphk isoforms have been identified: sphingosine kinase 1 (Sphk1) and 2 (Sphk2) [107,108]. Although the activity of both isoforms overlaps to some extent, they differ in substrate specificity, temporal expression patterns during development, and subcellular localization, suggesting their involvement in different cellular processes [109]. Both Sphk1 and Sphk2 are necessary for maintaining physiological levels of S1P, but they play essential yet antagonistic roles.

Under proinflammatory stimuli, Sphk1 translocates to the plasma membrane to generate S1P as an intracellular messenger [105]. Notably, Sphk1 stimulates cell growth and survival while suppressing apoptosis [110], and it is a key factor in maintaining cellular homeostasis under stress conditions by interacting with protein kinase R (PKR) [111].

Sphk1 is regulated “inside-out” by a complex array of post-transcriptional, epigenetics, and post-translacional mechanisms that ultimately generate a pool of S1P [112]. Its primary location is the cytosol, from where it translocates to the cell membrane through a process mediated by the Ca^2+^-myristoil switch protein calcium and integrin-binding protein. Once there, its activation takes place via phosphorylation of the Ser 225 by the action of the extracellular signal-regulated kinase (ERK) [113]. The activation of Sphk1 is necessary for its pro-proliferative and pro-survival signaling [114] and begins with the binding of anionic lipids such as phosphatidylserine (PS), phosphatidic acid (PA), and phosphatidylinositols (PIs) [115,116], as well as growth factors and cytokines [117,118].

In contrast to Sphk1, the functions of Sphk2 depend on its subcellular localization and cell type. Sphk2 promotes apoptosis in the ER, regulates mitochondrial respiration in mitochondria, modulates gene expression and telomere integrity in the nucleus, and is present in the plasma membrane in cancer [119].

#### 2.1.2. S1P Degradation Enzymes

The degradation of S1P is influenced by the cellular localization of specific enzymes involved in the process. S1P degradation occurs irreversibly due to the activity of SGPL1 and reversibly by two S1P phosphatases, SGPP1 and SGPP2 [120]. SGPL1 is located on the ER membrane, specifically facing the cytosolic side, and acts as an important regulator of S1P levels. This enzyme catalyzes the breakdown of S1P into hexadecenal and phosphoethanolamine, effectively reducing the intracellular concentration of S1P [121]. Previous studies have shown that loss-of-function mutations in SPGL1 result in the pathological accumulation of sphingolipid intermediates and lead to apoptosis induction [122].

On the other hand, SGPP1 is located on the ER membrane, facing the cytosolic side, while SGPP2 is located on the ER membrane, facing the luminal side [123,124]. Thereby, SGPP1 and SGPP2 can modulate S1P levels by catalyzing the dephosphorylation of S1P, converting it into sphingosine. These isoenzymes exhibit high specificity for sphingoid base phosphates, although they are expressed in different tissues [125]. The phosphorylation-dephosphorylation process allows for the dynamic regulation of S1P/CER and its downstream signaling.

Additionally, sphingosine can be converted back to CER by adding a fatty acid, catalyzed by ceramide synthase, constituting the sphingolipid recycling pathway [126]. The sphingolipid recycling pathway helps maintain the balance of sphingolipid levels and ensures the availability of ceramide for various cellular processes, including lipid signaling, membrane maintenance, and apoptosis regulation [127].

### 2.2. S1P Signaling

Although sphingolipids are well-known components of cell membranes, they also participate in both inter- and intracellular signaling. S1P, classified as an amphipathic lysophospholipid, possesses a polar head and a hydrophobic chain, allowing it to be released from the plasma membrane and act as an intracellular mediator and a ligand for S1P receptors (S1PR). Five isoforms have been described (S1PR1-5), belonging to the family of G-protein-coupled receptors (GPCR) coupled to αio, αq, or α12/13 proteins [128,129,130]. Consequently, critical signaling molecules such as phospholipase C (PLC), ERK, phosphoinositide 3-kinase (PI3K), and protein kinase B (Akt) are activated. Akt phosphorylates the third intracellular loop of S1PR1-2, leading to the stimulation of Rac, a member of the Rho family of GTPases [131]. These intracellular signaling events ultimately regulate processes such as angiogenesis, immunity, directed cell migration, proadhesion, and vascular permeability regulation during inflammatory responses in the endothelium, all of which are closely related to the scope of this review.

Numerous studies have demonstrated the crucial regulatory role of S1P in immune responses [93], where its primary function is to orchestrate the dynamic trafficking of lymphocytes and other immune cells, facilitating their migration into lymphoid tissue and subsequent egress to the blood [132,133]. In the CNS, S1P influences microglial and astrocytic activity and promotes vascular integrity. Furthermore, it is involved in the production of cytokines and chemokines through the indirect activation of TLR-4, contributing to the maintenance of cerebral homeostasis. The participation of S1P has also been demonstrated in inflammatory diseases including cancer and diabetes, in pathophysiological processes such as atherosclerosis and osteoporosis, and even in chronic diseases, disorders, and autoimmune diseases [134,135,136].

The knowledge about the functions of S1PRs is mainly based on the use (even at the clinical level) of S1PR modulators, such as fingolimod, ozanimod, siponimod, and others [137]. The mechanisms of action of these compounds may be complex, probably acting through multiple S1PRs and different intracellular signaling pathways [137,138]. Their classical mode of action is functional antagonism, based on internalization and subsequent degradation of S1PRs from the lymphocytes cell surface, preventing lymphocyte trafficking between the lymph node, blood, and CNS [139]. Furthermore, these compounds can cross the BBB and are capable of modulating the expression and signaling of S1PRs expressed on endothelial cells and brain parenchymal cellular populations [140].

S1PR1-5 are expressed by various subtypes of innate immune cells. While S1PR1 is expressed in most of these cells, S1PR2 is predominantly found in macrophages, eosinophils, mast cells, and monocytes [141]. Additionally, S1PR3 and S1PR4 are also expressed in neutrophils and dendritic cells [142]. Lastly, S1PR5 is present in circulating monocytes and NK cells [143]. The detailed implications of S1PRs in the immune response and barrier function will be further elucidated in the next section.

Figure 1 illustrates the metabolism and signaling of S1P.

## 3. S1P and the Immune System

As previously underscored, S1P signaling is essential in immune responses orchestrating the egress of lymphocytes from lymphoid tissues to blood. In the late 1990s, the new molecule FTY720 showed potent immunosuppressant activity leading to lymphopenia in rats [144] through a mechanism later identified as S1P-dependent [145], with S1PR1 playing a leading role [146]. These discoveries led to the emergence of FTY720, named fingolimod, as the first S1P-based drug approved for treating MS [100]. The proposed mechanism of action was, briefly, the retention of central memory T cells in lymph nodes by CC-chemokine receptor 7 (CCR7) after the aberrant internalization of S1PR1 caused by fingolimod. S1P levels are high in blood and lymph, helping immune cells to reach the vasculature and stabilizing the vessels. Fine-tuned gradients of S1P are critical for the exit and entrance of immune cells (not only T cells, but also B cells, NK cells mainly through S1PR5, and others) from primary and secondary lymphoid organs, and in nonlymphoid tissues. The precise mechanisms governing this traffic are not fully understood and the available evidence has been exhaustively reviewed both under physiological conditions [147] and after immune activation [148].

The expression patterns of S1PRs have emerged as key regulators in the inflammatory response. S1PR1 is almost ubiquitous being expressed in various cell types, including peripheral immune cells, endothelial cells, astrocytes, microglia, neurons, and to a lesser extent, oligodendrocytes [149]. Activation of S1PR1 on myeloid cells promotes neuroinflammation [150]. For its part, S1PR2 is widely expressed in various immune cells, including T cells, B cells, dendritic cells, macrophages, and natural killer (NK) cells, indicating its involvement in modulating immune responses and inflammatory processes. While S1PR2 predominates in proinflammatory cells, S1PR1 is also upregulated during the resolution phase of inflammation, facilitating macrophage migration from the inflammatory site and promoting the resolution process [151]. Both S1PR1 and S1PR2 signaling seem crucial for establishing and maintaining endothelial barrier function [152,153]. However, controversial data point in the opposite direction under specific circumstances, as discussed later.

S1PR3 is expressed in various cells and tissues, including endothelial cells, smooth muscle cells, neurons, glial cells, and cells of the immune system, such as T lymphocytes and dendritic cells. Under proinflammatory conditions, it can promote the migration of mature dendritic cells and mediate the chemotaxis of macrophages, neutrophils, and monocytes, driving leukocyte movement and recruitment to the site of inflammation [154]. Additionally, S1PR3 induces bactericidal action in macrophages through the production of reactive oxygen species (ROS), thereby enhancing the immune response [155].

S1PR4 is specifically expressed in lymphoid tissues and immune cells, including lymphocytes, dendritic cells, and macrophages. Its sole expression in these cell types suggests its relevance in immune responses and inflammatory processes, as it has been associated with cancer, autoimmune diseases, and IBD [92]. However, there is limited and controversial information about the mechanisms controlled by S1PR4. S1PR4 may participate in the release of proinflammatory cytokines in activated macrophages and mediate the activation and maturation of dendritic cells [156], but also its absence exacerbates M1 polarization and pulmonary inflammation [157].

S1PR5 is expressed in NK cells and patrolling monocytes. Similarly to S1PR4 and despite being one of the ozanimod targets, S1PR5 has been less studied. Available evidence points to its role in regulating NK cell migration, guiding their exit from the bone marrow and facilitating their circulation in the bloodstream [158], as well as in the egress of monocytes from the bone marrow and the inhibition of phagocytosis [159].

Beyond its peripheral participation in the immune cell migration toward the vasculature, S1P also regulates inflammation and immune events in the CNS. Most cell types in the CNS (neurons, microglia, astrocytes, oligodendrocytes, and BBB cells) express S1P receptors, and different S1P-based drugs have shown direct effects on them [160,161]. S1P, a lipid that easily accesses the CNS through the BBB, is the most enriched lipid in the CNS and plays an indubitable role in development. However, the neurotoxic/neuroprotective actions of S1P are still under debate, especially under disease conditions [162]. Some of the identified S1P pathways in the CNS are related to immune modulation, particularly, neuroinflammation. Glial activation is one of the main features of neuroinflammation and S1P may drive it. S1P accumulation has been shown to activate microglia in neural SPGL1 ablated mice through S1PR2 [163], in the microenvironment of degenerated intervertebral discs [164], and in glioma progression [165]. Both fingolimod and siponimods boost the expression of anti-inflammatory phenotypes and genes in microglia [166,167] and astrocytes [168]. Overexpression of S1PR1 is present in reactive astrocytes after fingolimod discontinuation [169] whereas this drug inhibits astrogliosis and its associated neuroinflammation in mice [170]. S1PRs are also involved in CXCL1 release from astrocytes [171] and CXCL5 from astrocytes and microglia after TLR4 stimulation [172]. All S1PR1-5 receptors, except S1PR4, have demonstrated participation in the activation of microglia and/or astrocytes, leading to neuroinflammation [173]. Nevertheless, their signaling is complex, as previously highlighted, and there are still controversial data that fuel an active debate about their exact neuroimmune mechanisms.

Beyond receptors, different pro/anti-inflammatory effects have been associated with metabolic S1P enzymes. Sphk1 overexpression plays a crucial role in the development of inflammatory and immune-related diseases, such as inflammatory bowel disease, Alzheimer’s disease, or hypertension. As a result, the antagonization of Sphk1 by Sphk1 or Sphk1/2 inhibitors is under investigation as a novel therapeutical alternative [174]. On the other hand, Sphk2 produces its own pool of S1P depending on its subcellular localization, which has been implicated in protection against ischemic injury, macrophage polarization, or regulation of cytokine expression [115]. However, there is scarce and controversial information on CNS diseases. Although the exact mechanisms behind the cytosolic/nuclear shift of Sphk2 remain elusive, a study suggests a neuroprotective activity of cytosolic Sphk2 in Alzheimer’s disease, as there is an inverse correlation between its cytosolic expression and amyloid deposits in the frontal cortex and hippocampus of AD patients, coinciding with translocation to the nucleus [114].

Consistent with the previously discussed functions of S1P, SGPL1 activity has been mostly associated with beneficial effects due to its anti-inflammatory properties [163,175,176,177] but it is also linked to deleterious effects on barrier function [178]. This ambivalent role depends on the cell type [179]. Similarly, the expression pattern of SGPP1 and SGPP2 in specific cell types may contribute differentially to the inflammatory context. SGPP2 activity is primarily upregulated during inflammation in many cells, such as endothelial cells and neutrophils [124]. In IBD models, SGPP2^−/−^ mice showed less severe dextran sodium sulfate (DSS)-induced colitis together with the suppression of inflammation and intestinal cell apoptosis, leading to a healthier mucosal barrier. On the other hand, SGPP1 deletion implies a higher proinflammatory response after DSS-induced colitis [180].

The significance of S1P in bridging the activity of the innate and adaptive immune systems, coupled with its modulation of barrier functions discussed below, renders it a compelling target for controlling inflammation and the immune response in pathologies associated with immune dysfunction and barrier permeability, such as psychiatric diseases and IBD. Figure 2 presents an integrated view of the involvement of S1P signaling and metabolism molecules in these processes.

## 4. S1P Signaling in Blood–Brain and Intestinal Barrier Functions

### 4.1. S1P Signaling in BBB Function

Increasing preclinical (in vitro and in vivo) and clinical evidence supports the regulatory role of S1P signaling pathways on the structure and function of the BBB in pathophysiological conditions, especially in neurological pathologies such as stroke or traumatic brain injury (TBI) [181].

S1P possesses three carrier proteins: albumin, apolipoprotein M (ApoM), and apolipoprotein A4 (ApoA4) [182]. The specific binding of S1P to these carriers could differentially affect S1P release and signaling [182]. Thus, several findings have shown that ApoM-bound S1P regulates BBB paracellular permeability and vesicle-mediated transport in mice brain vasculature [183], and high-density lipoprotein (HDL)-S1P is more effective in enhancing endothelial barrier function via S1PR1 compared to albumin-bound S1P [184].

The five G-protein-coupled S1P receptors (S1PR1–5) have selective expression profiles in the cellular types constituting the BBB and in those within the brain parenchyma (neurons, oligodendrocytes, and microglia), which can also influence BBB function [149]. S1PR1 stimulation in endothelium leads to the activation of the Rac pathway, enhancing barrier function by increasing cadherin distribution to the membrane and strengthening adherens junctions, TJs, and cytoskeletal stabilization [181,185,186]. In this vein, several studies have demonstrated the ability of fingolimod to reduce basal P-glycoprotein activity in isolated rat brain capillaries through a mechanism related to the S1P transporter multidrug resistance-associated protein 1 (Mrp1) [187,188]. P-glycoprotein is an ATP-driven drug efflux pump that prevents the entry of drugs across the BBB. Fingolimod and other S1P modulators could be attractive tools to improve drug delivery to the brain. The transporters Mrp1 and Spinster homolog 2 (Spns2) are essential for the effects of S1P on BBB integrity, ensuring a proper bioavailability of S1P in the vascular niche [189].

In a complementary mechanism, recent studies suggest that CYM-5442, a novel and selective modulator of S1PR1, maintains the integrity of the BBB by restricting vesicle transcytosis after TBI and during acute ischemic stroke in rodents [190,191].

The protective role of these compounds in neurological/neurodegenerative diseases may also be related to their S1PR1-dependent immunomodulatory actions on parenchymal glia and neurons [192,193,194].

Some authors have reported controversial data regarding the role of S1PR1 in the pathophysiology of stroke. Specifically, Gaire et al. found that the oral administration of AUY954, a selective functional antagonist of S1PR1, ameliorates brain infarction, neurological deficit score, and neural cell death in a transient focal cerebral ischemia model in rats [195]. Similarly, Mandeville et al. reported that fingolimod does not reduce infarction, brain swelling, hemorrhagic transformation, and behavioral outcome after focal cerebral ischemia in mice [196]. Further scientific efforts are needed to disentangle the precise pharmacological profile and intracellular signaling pathways activated by the different S1P modulators in the widest possible range of experimental conditions.

On the contrary, the isoform S1PR2 in endothelium has detrimental effects, increasing BBB permeability, oxidative stress, and stimulating inflammation and subsequent leukocyte trafficking in several pathological scenarios [197,198]. Genetic or pharmacological inactivation of S1PR2 prevents BBB disruption and microgliosis in the experimental autoimmune encephalomyelitis (EAE) animal model of MS [199], and microglial activation and M1 polarization following experimental cerebral ischemia through an ERK1/2 and JNK-related mechanism [200]. The expression of S1PR2 in pericytes is regulated by microRNA-149-5p in a crucial mechanism related to the increase in BBB permeability produced after transient middle cerebral artery occlusion in rats [201]. It is worth noting that there is the sexually dimorphic expression of S1PR2 in the CNS [202], which underlies the increased female susceptibility to BBB dysfunction and worsened symptomatology during autoimmune diseases [203].

Increasing evidence suggests the role of S1PR3 as a pathogenic mediator in certain experimental conditions. Thus, the upregulation of S1PR3 and activation of RhoA-related signaling in astrocytes induce inflammation [204], and its activation can promote endothelial cell contraction, disrupting vascular barriers [205]. Furthermore, the administration of CAY10444, an S1PR3 antagonist, reduces BBB injury via the downregulation of the C-C motif chemokine ligand 2—C-C motif chemokine receptor 2 (CCL2-CCR2) axis, p-p38 mitogen-activated protein kinase (MAPK), and intercellular adhesion molecule 1 (ICAM1), and the upregulation of ZO-1 following acute intracerebral hemorrhage in in vivo and in vitro rat models [206]. CAY10444 also prevents M1 microglial activation after transient focal cerebral ischemia through a mechanism related to the phosphorylation of ERK1/2, p38 MAPK, and Akt [207].

S1PR4 is mainly expressed in immune cells, but there may exist some expression in the brain. Recently, an overall barrier-protective function of endothelial S1PR4 receptor has been suggested both in vitro and in vivo [208].

Seminal studies showed that the activation of S1PR5 with a selective agonist (azacyclic analogue of FTY720) promotes BBB integrity and reduces inflammation and transendothelial migration of monocytes in cultures of human brain endothelial cells [209].

Ozanimod (RPC-1063) is a newly developed compound that can selectively modulate S1PR1/5, reducing the dysfunction of the BBB and exerting neuroprotective actions following intracerebral hemorrhage in mice [210]. Similarly, the use of A-971432, a selective S1PR5 agonist, preserves BBB integrity and activates pro-survival pathways, such as brain-derived neurotrophic factor (BDNF), Akt, and ERK, in an animal model of Huntington’s disease [211].

Another approach to studying the effect of S1P on the BBB is the genetic or pharmacological modulation of other members of the S1P signaling besides receptors. Thus, Sphk1 has been involved in endocytic membrane trafficking, ensuring the conversion of sphingosine to S1P [212]. The other isoform, Sphk2, is necessary to regulate junctional protein expression and BBB protection in hypoxic preconditioning-induced cerebral ischemic tolerance in rats [213]. The histone methyltransferase Smyd2-dependent methylation of the Sphk/S1PR signaling pathway produces BBB disruption in experimental ischemic stroke in rats [214]. The preservation of S1P levels produced by the downregulation of SGPL1 enhances barrier function in human cerebral microvascular endothelial cells following an inflammatory challenge [178]. Similarly, the administration of fingolimod reduces SGPL1, increasing the levels of S1P and reversing BBB leakiness during EAE [215].

### 4.2. S1P Signaling in Intestinal Barrier Function

As previously outlined, the interaction between S1P and its receptor S1PR1 regulates lymphocyte trafficking from the spleen and lymph nodes into the systemic circulation. Modulation of S1PR1 leads to the reversible sequestration of specific lymphocyte subsets in lymph nodes, resulting in decreased peripheral circulating lymphocytes and reduced tissue inflammatory infiltrates. While the direct relationship between the S1P pathway and intestinal barrier function in vivo lacks sufficient evidence, some experimental data suggest its possibility, although with controversial results.

For instance, some studies have demonstrated that S1P can increase levels of E-cadherin, both in cellular amounts and at the cell-to-cell junctions, promoting improved barrier integrity in cultured IECs [216]. Additionally, S1P signaling through the S1PR2 receptor and Rho kinase (ROCK) has been implicated in IEC extrusion, a crucial phenomenon for maintaining homeostatic cell numbers in the intestine and, consequently, effective barrier function under physiological circumstances [217]. This signaling pathway has also been associated with interactions between IECs and T cells during experimental colitis [218].

Furthermore, treatment with a selective S1PR1 agonist has been shown to enhance intestinal barrier function in IL-10^−/−^ mice, a well-recognized model of chronic colitis, as evinced by restoration of TJ protein expression (occludin and ZO-1) and suppression of epithelial cell apoptosis [219]. Taking advantage of the Sphk2^−/−^ mice, our group has recently shown that experimental stress induces colonic inflammation by modulating S1P pathways—it increases S1P in the colon possibly due to a downregulation of Sphk2 and its degradation enzymes—leading to dysregulated innate and adaptive immune responses and enhanced intestinal permeability. Moreover, Sphk2^−/−^ mice presented a cytokine-expression profile towards a boosted Th17 response, lower expression of claudins, and structural abnormalities in the colon [88].

Further research is currently underway to understand the role of the S1P pathway in endothelial barrier function in the gut. It is known that the S1PR1 receptor is expressed by endothelial cells in the intestinal vasculature [220,221], and its genetic deletion in mice increases colonic vascular permeability [220]. Continued investigations are necessary to fully elucidate the intricate mechanisms and implications of S1P signaling in maintaining intestinal barrier function in health and disease.

## 5. Evidence of S1P Role in Psychiatric Diseases

### 5.1. S1P in MDD

MDD is one of the first sources of disability in terms of Disability Adjusted Life Years (DALYs) and some projections foresee that it will be the main cause of disability and one of the leading sources of morbidity by the year 2030 [222]. Importantly, these severe predictions are prior to the COVID-19 scenario, in which the levels of stress and its possible effects on mental health are still under scrutiny.

The etiology of MDD is still not fully understood. However, one thing that seems clear is that its pathophysiology goes far beyond the classical monoaminergic hypotheses involving neurotransmitter imbalances of serotonin (5-HT), noradrenaline (NA), and dopamine (DA). Inflammatory processes have been linked to MDD development and depressive-like symptoms [30], with patients frequently displaying high levels of immune dysfunction biomarkers, and inflammation being associated with the clinical severity of mood disorders [223,224]. The relationship between the immune system and MDD raises questions about its origin and mechanisms and S1P emerges as a molecule worth studying, given its proinflammatory actions through immune cell recruitment [129].

One relevant process is to identify how peripheral inflammatory parameters with an important size, such as cytokines, can infiltrate the CNS through the BBB. Even more, bacteria from the own patients’ microbiota have been alluded to as inductors of the inflammatory response in MDD as a consequence of the malfunction of some barriers, such as the intestinal barrier, in an already mentioned phenomenon called leaky gut [84,85]. S1P plays a critical role in upholding both the integrity of the BBB and the intestinal barrier, making them a plausible link to MDD [225].

Animal models, mainly using stress-based models and particularly chronic mild stress (CMS), have shed light on the potential role of the S1P pathways in MDD pathophysiology. These models are well-described and commonly employed to investigate MDD pathological basis and screen for new antidepressants [226]. In a recent article, Guo et al. [227] have shown that the pharmacological blockage of S1PRs in the CMS model alleviated depressive-like behavior, hippocampal damage, inflammation, and oxidative stress due to NLRP3 inflammasome activation, highlighting the involvement of the S1P pathway, particularly its actions on the immune system, in MDD etiology. 

A study investigating the connection between gut microbiome-derived lactate and anxiety-like behaviors in chronically stressed rats has found that S1PR2 protein expression in the hippocampus is lower in stressed rats, negatively correlating with symptom severity [228]. Furthermore, this study shows that S1PR2 reduces the TNF-α increased levels after stress exposure, suggesting the potential role of S1PR2 in mediating stress-related psychiatric phenotypes.

Another research has also focused on the microbiota and its impact on depressive-like behavior, but in this case in the possible translocation of oral microbiota through the BBB, using a preclinical model combining periodontitis and CMS in rats. This study identified variations in the expression of significant mediators involved in the BBB integrity, neuroinflammatory parameters, and S1P signaling modulation in the frontal cortex [229]. Once more, these results suggest a connection between the S1P pathway and MDD, particularly through its actions on BBB integrity and the immune response.

In a significant study including animal and patient samples, it has been revealed that experimental overexpression of S1PR3 in the medial prefrontal cortex (mPFC) led to a resilient phenotype, whereas knock-down of S1PR3 resulted in vulnerability with higher anxiety- and depressive-like behaviors, effects mediated, once again, by TNF-α [230]. Importantly, this study also observed decreased S1PR3 mRNA expression in the blood of war veterans with posttraumatic stress disorder (PTSD), and its expression negatively correlated with symptom severity. All in all, this research strongly suggests that S1PR3 might be a regulator of stress resilience and underscore sphingolipid receptors as valuable substrates of importance to stress-related psychiatric disorders.

The existence of patients who are refractory to antidepressant treatments is a significant concern in MDD research. Thus, it is crucial to examine factors that regulate recovery. In this sense and linking once again TNF-α and the S1P pathway in depression models, enhancing endothelial barrier integrity through S1PR stimulation or TNF-α inhibition contributes to the recovery from prolonged learned helplessness depression-like behavior in mice [231].

In summary, the vast majority of studies about S1P and depression have been conducted in animal models. While there are relatively few articles about the role of S1P in MDD, all available evidence points towards a connection between MDD and S1P through its proinflammatory actions and its role in maintaining the integrity of various physiological barriers. Particularly, there appears to be a clear link between the actions of TNF-α, a proinflammatory cytokine increased in both animal models and patients with MDD [232], and the S1P pathway. Moreover, connections among other immune parameters involved in neuroinflammation and barrier integrity, and S1P are rather likely.

### 5.2. S1P in Neurodevelopmental Disorders

Neurodevelopment encompasses the proliferation and maturation of neural cells to establish proper neural circuits for vegetative and cognitive functions. This phenomenon involves processes like cell migration, plasticity, metabolic changes, and myelination, specific to certain cells and time windows in the CNS [233]. Neurodevelopment requires a controlled microenvironment, where glial cells and specifically microglia play crucial roles [234,235]. Disturbances during this process increase the risk for neurodevelopmental disorders, including SZ, ASD, and attention deficit/hyperactive disorder (ADHD) [236].

#### 5.2.1. S1P in SZ

SZ is a severe mental disorder with a multifactorial etiology characterized by psychotic episodes that usually emerge during late adolescence or early adulthood [237]. The SZ pathophysiology may involve S1P signaling in four main aspects: myelination, neurotransmitter release, synaptic pruning, and BBB function. The developmental risk factor model for SZ suggests that genetic disturbances, mainly related to innate immunity and synaptic proteins, coupled with environmental insults, such as perinatal stress or cannabis exposure, interact for the onset of the disease. The establishment of neural ensembles, functional units of the CNS, relies on the adequate structure and function of the cells. Myelination is crucial for the optimal conduction of nerve signals, and oligodendrocytes oversee this process in the CNS. The loss of white matter (rich in myelin) is a consistent finding in SZ [238], suggesting potential vulnerability and dysfunction of oligodendrocytes during neurodevelopment, which may increase SZ risk [239]. S1P signaling plays a fundamental role in myelin production [240], and alterations in S1P could be associated with white matter changes in SZ. Studies in patients have reported decreased S1P levels and a ratio imbalance in favor of S1P precursors, sphingosine and ceramide, in the corpus callosum, which may impact on apoptosis and cell-cycle arrest [241]. Additionally, increased transcript levels of S1P receptors have been observed [242].

Excessive synaptic pruning during neurodevelopment is another disruption linked to the increased risk of SZ. The complement system, a part of innate immunity, plays a critical role in synapse elimination and, consequently, in neural circuit maturation. Genetic variations in the complement system have been associated with SZ, leading to overexpression of complement component 4 (C4), oxidative/nitrosative stress, and excessive synaptic pruning in corticolimbic regions [242,243,244]. Synaptic bouton elimination depends on labeling by complement components C1q and C3. C1q/C3 complex interacts with the C3 receptor (C3R), highly expressed in microglia, and these cells engulf portions of the synaptic bouton in a phagocytosis-related phenomenon [245]. Complement labeling on synapses depends on various factors, including weak or atrophied synapses [246]. Although the specific role of S1P in the pruning hypothesis of SZ is not well understood, existing evidence suggests that S1P may be involved in this process. S1P is essential for glutamate exocytosis in presynaptic terminals by modulating synapsin I distribution [247], and decreased synapsins have been detected in cortical regions in SZ [248]. Thus, lower levels of S1P might be associated with impaired glutamatergic transmission and weak synapses, increasing the likelihood of elimination. S1P has also been implicated in dopamine release in rodents [249], potentially influencing mesolimbic dopaminergic tone and the core clinical feature of SZ, psychosis [250]. These findings suggest that S1P and its receptors could be potential therapeutic targets for SZ.

Astrocytes, which are part of the BBB and rich in Sphks and S1PR [206,251], have been reported with varied density, activity states, and genomics in SZ [252]. These cells are sensitive to inflammatory stimuli, and there are multiple subtypes of astrocytes with different characteristics and functions depending on the brain region [253]. These features can explain the heterogenous information about astrocytes in SZ. Studies have reported increased BBB permeability in temporal lobe structures, including the hippocampus, in postmortem samples of SZ [254]. As S1P modulates BBB permeability [181], further investigations are needed to explore the potential relationship between S1P and BBB dysfunction in SZ.

#### 5.2.2. S1P in Autism Spectrum Disorders

Autism spectrum disorders (ASD) are another neurodevelopmental disorder with a strong genetic basis, characterized by abnormal brain development and a range of behavioral characteristics, including social isolation, language and motor skill delays, impulsiveness, hyperactivity, and sometimes seizures, irritability, and agitation [255].

S1P has been proposed as a biomarker for aiding in ASD diagnosis and disease status, as serum S1P levels are reduced in patients compared to controls [256]. However, S1P levels seem unaltered in the prefrontal cortex, a region with impaired function in ASD [257]. Additionally, single nucleotide polymorphisms (SNPs) in the *Sphk1* gene have been associated with ASD in females [258].

Animal models of ASD have shown lower hippocampal and serum S1P and SphK levels together with memory impairments in rats exposed to valproic acid during gestational age [259,260]. Sphk/S1P-related pathways may also be related to dendritic spine pathology in the corticolimbic system, a core pathophysiological characteristic in ASD [261].

#### 5.2.3. S1P in Attention Deficit Hyperactivity Disorder

Attention deficit hyperactivity disorder (ADHD) is a mental disorder characterized by symptoms of lack of attention, impulsivity, and hyperactivity, typically observed from childhood to adulthood. Multiple neurotransmitter systems, including the dopaminergic and noradrenergic, are dysfunctional in ADHD [262]. Furthermore, studies in patients diagnosed with ADHD have reported genetic alterations related to ceramide synthesis and, consequently, to S1P [263]. Interestingly, plasma levels of sphingomyelins were found to be reduced in children with ADHD [264], while S1P plasma levels were found to be upregulated in adults [265]. These findings suggest that there may be dynamic alterations in ceramide metabolism throughout the course of the disease and highlight the potential role of these molecules as potential biomarkers for diagnosing and assessing disease status in ADHD.

## 6. Current and Potential Drugs Targeting S1P in Psychiatric Diseases and IBD

Several compounds differentially target the isoforms of the S1PRs, having all in common the functional antagonism of S1PR1 in immune cells through its internalization. Among these drugs, fingolimod, siponimod, ozanimod, and etrasimod have garnered attention in this review due to their actions. Fingolimod was the pioneering development, displaying a broader receptor affinity with effects on S1PR1, 3, 4, and 5. Subsequent compound designs have emphasized enhancing specificity while minimizing adverse side effects. Siponimod selectively targets S1PR1 and S1PR5, avoiding the activation of S1PR3 [266], and ozanimod was developed later as a potent agonist of S1PR1 and 5, with residual activity over the S1PR2,3, and 4 [267]. On the other hand, etrasimod acts as a full agonist of S1PR1 and a partial agonist of S1PR4 and 5 [268]. Figure 3 illustrates the activity of S1PR modulators on the different isoforms of the S1PRs summarizing the evidence gathered in this review relevant to psychiatric diseases and IBD.

The primary aim of this section is to review the potential pharmacological applications of targeting S1P in the field of psychiatry. Currently, the previously mentioned S1PR modulators (fingolimod, siponimod, ozanimod, etrasimod) together with ponesimod are commercially available drugs for treating MS (a CNS pathology), while ozanimod for managing ulcerative colitis (an intestinal disease). As a result, we have opted to include pertinent information about S1P drugs in IBD, not only due to the approval of ozanimod but also because of the previously emphasized significance of intestinal barriers in psychiatric diseases.

### 6.1. S1P-Related Drugs for the Treatment of Psychiatric Diseases

#### 6.1.1. S1P-Related Drugs for MDD

This is a relatively new area of research, and consequently, there are limited studies concentrating on the effects of pharmacological modulation of the S1P pathway on the symptomatology and clinical outcome of MDD. The main rationale for considering S1P-related drugs as potential treatments for MDD lies in the presumed role of inflammation in the MDD pathophysiology and the importance of the S1P pathway in maintaining barrier integrity, as previously explained. Therefore, most studies covering this subject have been conducted using animal models, with a particular focus on the immune response.

Many of these studies employ fingolimod, the first S1PR modulator approved for treating MS. A previously mentioned recent study has provided theoretical support for S1P receptor modulation in MDD treatment, demonstrating that fingolimod protects hippocampal neurons from CMS-induced damage and lessens depressive-like behavior by inhibiting neuroinflammation [227]. The improved symptomatology observed in the CMS rats in this research was attributed to blocking the NLRP3 inflammasome in hippocampal microglia and polarizing them to the M2 phenotype.

Another study employing a validated neuropsychiatric lupus model in mice has indicated that fingolimod significantly attenuates depression-like behavior, leading to reduced brain T cell and macrophage infiltration, and a significant decrease in cortical leakage of serum albumin [269]. Additionally, astrocytes and endothelial cells from fingolimod-treated mice showed lower expression of inflammatory genes. These findings further indicate the potential of S1P signaling modulation as a novel therapeutic target for depressive symptoms, particularly in the context of neuropsychiatric manifestations in patients with lupus erythematosus.

While there are no specific studies examining the potential antidepressant effects of fingolimod in patients with MDD, studies in patients with MS provide some evidence in support of this beneficial action, as fingolimod is indicated for the treatment of MS and patients with MS are at higher risk for MDD. Notably, studies investigating the fingolimod effects on depressive symptoms in MS patients have shown an improvement in depressive symptoms compared to other treatments for relapsing-remitting MS [270], underscoring the potential antidepressant impact of S1P pathway modulation.

Other potential pharmacological approaches exist, although they have yet to be extensively tested in patients with MDD. This would be the case with the functional inhibitors of the acid sphingomyelinase (ASM), an enzyme whose activity is increased in patients with MDD according to a pilot study [271]. ASM is a lipid metabolizing enzyme that can be activated by proinflammatory cytokines (e.g., TNF-α, IL-1β, IFN-α), leading to the cleavage of sphingomyelin to ceramide, a precursor of S1P. Hence, ASM inhibition has been proposed as a therapeutic approach for several pathologies due to its potential antiapoptotic and neuroprotective effects [272].

#### 6.1.2. S1P-Related Drugs for Neurodevelopmental Disorders

The pharmacological modulation of S1P-related pathways holds great promise as a potential treatment for SZ. A recent study has shown that fingolimod exerts protective effects on behavioral and inflammatory alterations induced by short-term exposure to cuprizone, a novel psychosis mode [273]. Specifically, fingolimod prevented the activation of microglia, cytokine release, and infiltration of leukocytes in the CNS while alleviating methamphetamine hypersensitivity.

Fingolimod has also demonstrated beneficial effects on white matter [274,275], improving white matter microstructure in an EAE rodent model [276] and in patients with impaired pyramidal function [277]. Also, in vitro studies have further shown that fingolimod promotes oligodendrocyte survival [278] and proliferation [279], as well as enhances precursor cell mitogenesis and re-myelination [280,281]. These findings collectively suggest a hypothetical mechanism for fingolimod in maintaining white matter integrity in SZ [282] and ASD, given that impaired uncinate fasciculus (axons connecting limbic areas of the temporal lobe structures with the prefrontal cortex) [283] and superficial white matter development are consistent findings in ASD [284].

Fingolimod may also have a potentially beneficial effect on neurodevelopmental disorders through its ability to increase BDNF [285], which can even stimulate spinogenesis in vitro [286]. This effect can be relevant for diseases with dendritic spine pathology as a core mechanism, such as SZ and ASD [287]. However, a preliminary study in patients diagnosed with Rett syndrome, a genetic neurodevelopmental disorder that compromises the adequate maturation of the nervous system, did not show improvement in symptoms or metabolic and molecular markers with fingolimod treatment [285].

The SphK blocker SKI-II has shown promising effects in attenuating memory impairments and stimulating hippocampal synaptic plasticity, autophagy, and cell surveillance-related pathways in the valproic acid-induced ASD rat model [259]. Furthermore, SKI-II has recovered behavioral and brain molecular impairments in BTBR T^+^ Itpr3^tf^/J mice [288], a strain commonly used to study ASD due to its Disc 1 (disrupted in SZ 1) gene deletion, which is associated with an increased risk not only for ASD but also for multiple mental diseases. These data suggest that the Sphk/S1P pathway could be a potential novel target for ASD pharmacotherapy.

Table 1 presents a comprehensive compilation of both preclinical and clinical studies providing compelling evidence of the potential involvement of S1P in psychiatric diseases.

### 6.2. S1P-Related Drugs for IBD

As previously highlighted, enhanced leukocyte recruitment within the gut represents a key pathophysiological event in IBD, and blockade of T lymphocyte trafficking—for example with a monoclonal antibody directed against α4β7 integrin such as vedolizumab—is a widely used therapy for both ulcerative colitis (UC) and Crohn’s disease (CD). In this context, S1P pathway is now emerging as a promising target for human IBD treatment [289] since its pharmacological modulation leads to the internalization of S1PR1 receptors in lymphocytes, preventing their mobilization from lymph nodes. This S1PR1 internalization causes lymphocyte subpopulations to be sequestered in the aforementioned lymphoid organs, preventing their circulation and recruitment to inflamed tissues, such as the gut. Experimental evidence suggests that targeting S1P may be beneficial in treating intestinal inflammation [219]. Additionally, S1PR1 receptor genetic deletion in mice has been shown to increase bleeding in experimental colitis induced by oral administration of DSS [220]. However, the anti-inflammatory properties of S1P receptor agonists in experimental colitis may not solely be due to their effects on lymphocyte trafficking, but also to potential effects on dendritic cell migration and, as previously mentioned, vascular barrier function [221]. This event, subsequently, leads to a diminished number of circulating and tissue lymphocytes.

This fact is supported by promising results of phase II and III clinical trials targeting S1P receptor subtypes 1 and 5; up to date, two S1P receptor modulators, ozanimod and etrasimod, have been tested for IBD. A first placebo-controlled, phase II trial that included 197 adult patients with moderate to severe UC showed that 1 mg per day of ozanimod—a molecule with high affinity to S1P receptor subtypes 1 and 5—resulted in a slightly higher rate of clinical response and remission at week 8 and 32 than placebo [290]. In a long-term, open-label extension of this study, clinical, endoscopic, and histological benefits with ozanimod were observed after two years of treatment, with a favorable safety profile after 4 years of follow-up [291]. A subsequent randomized, placebo-controlled, phase III trial published in 2021, analyzed the effect of ozanimod as induction and maintenance therapy for patients with moderate to severe UC, including more than 1000 patients for the induction trial and 457 in the maintenance study. The results demonstrated that the rate of clinical remission and response was significantly higher among patients who received ozanimod compared to those who received a placebo during both induction and maintenance periods. [292]. Following these results, ozanimod became the first S1P receptor modulator approved by the U.S. Food and Drug Administration and the European Medicines Agency for the treatment of UC. Ongoing clinical trials are evaluating the efficacy of this drug in the treatment of CD.

Furthermore, two independent randomized, placebo-controlled, phase III trials recently published confirmed that etrasimod, another oral S1P receptor modulator that selectively activates S1PR1, 4, and 5, was effective in achieving clinical remission in both induction (12 weeks) and maintenance (52 weeks) periods in patients with moderately to severely active UC [293]. Importantly, the incidence of infectious adverse events and malignancies with ozanimod and etrasimod were similar to that with placebo. This finding could be related to data from a recent study suggesting that etrasimod reduced circulating levels of specific subsets of adaptive immune cells (T and B lymphocytes) without significant effects on innate immune cells, such as NK cells and monocytes, which are involved in immune surveillance [294]. It is also known that modulation of S1PR2 and S1PR3—but not S1PR1, 4, and 5—has been associated with other serious adverse events, including reduced pulmonary function, malignancies, macular oedema, and cardiac arrhythmias [295].

In alignment with Table 1, an array of preclinical and clinical evidence reinforcing the pivotal role of S1P in the context of IBD is assembled in Table 2.

## 7. Future Directions

This review has examined the current evidence demonstrating the ability of S1P to modulate the immune system and barrier functions (BBB and intestinal), processes that may underlie relevant pathophysiological features to psychiatric diseases. S1P signaling has emerged as a promising pharmacological target, particularly due to the proven efficacy of approved drugs in treating CNS and intestinal pathologies. So, is S1P at the helm of immune and barrier mechanisms in psychiatric diseases? It is not yet possible to give a conclusive response without first elucidating some critical points through prospective studies.

Addressing these questions poses several difficulties. Firstly, the exact molecular mechanisms underlying S1P signaling are unknown despite numerous research efforts. Although S1P is generally recognized to increase immune system activity and maintain barrier function, controversial data hinder a conclusive understanding of the role of some S1P elements due to complex signaling involving specific cell types and intricate intracellular and extracellular pathways. For instance, S1PR1 stimulation in the context of psychiatric diseases can be seen as deleterious for increasing inflammation but beneficial for promoting barrier integrity. More mechanistic studies are mandatory to understand the molecular functioning of all the S1P pathway elements and explain the big picture.

Secondly, preclinical and clinical evidence primarily relies on nonspecific S1PR modulators, making it challenging to attribute their effects to specific isoforms or cell types. New pharmacological tools specific to each S1P element could help to unravel the complex network signaling of S1P. Despite this limitation, the significant studies gathered in this review show promising actions of available S1P-based drugs in psychiatric diseases, demonstrating improvements in behavioral and pathophysiological features. While S1P clinical evidence is still limited, it reinforces the idea of S1P playing a role in these illnesses.

The final main difficulty is inherent to the nature of MDD, SZ, or ASD as multifactorial and heterogeneous diseases with elusive pathophysiology. Neuroinflammation and barrier alterations are transversal to all these pathologies, but they may merely be coincidental phenomena. If this were the case, the possibilities of S1P as a pharmacological alternative would decrease unless it modulates other essential mechanisms beyond immune and barrier functions. Moreover, relevant differences exist among psychiatric illnesses and even their subtypes. Therefore, future research should carefully analyze which pathologies or subtypes would benefit from controlling inflammation and barrier integrity to achieve clinical improvement.

In light of all the exposed reflections, the question in the title remains open but highly relevant. While a definite role of S1P in psychiatric diseases cannot be ascribed yet, emerging evidence, particularly in the last years, augurs a necessary intensification of research in this field. Targeting S1P has the potential to modulate immune and barrier alterations associated with these diseases and may have implications in their possible origin through the leaky gut phenomenon. As our understanding of the intricate signaling pathways and molecular mechanisms involving S1P continues to expand, so does the potential to target this lipid mediator for more effective and specific treatments in various psychiatric conditions. Innovative strategies, such as selective modulators of S1P receptors, combination therapies, and personalized medicine approaches, may usher in a new era of precision psychiatry. While challenges lie ahead, the ongoing pursuit of S1P-related research holds immense promise in alleviating the burden of psychiatric diseases and improving the lives of millions worldwide.

## Figures and Tables

**Figure 1 ijms-24-12634-f001:**
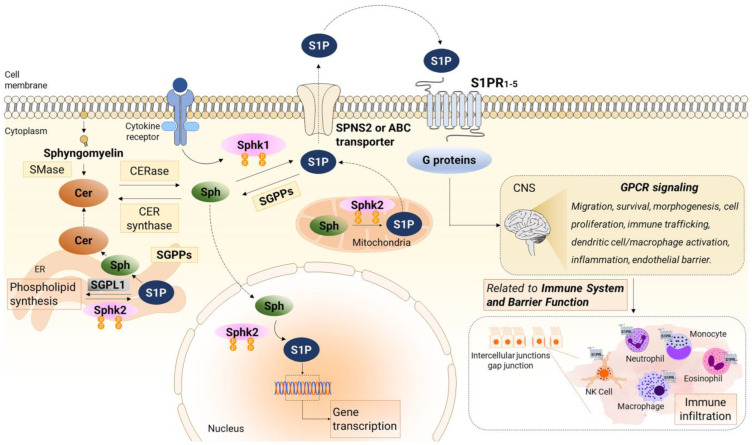
Sphingosine-1-phosphate (S1P) signaling and metabolism. Sphingosine kinase 1 (Sphk1) primarily resides in the cytosol and gets activated by cytokines, leading to the phosphorylation of Sphingosine (Sph) to generate S1P at the plasma membrane. Additionally, Sphingosine kinase 2 (Sphk2) phosphorylates Sph to produce S1P at various intracellular locations, such as the endoplasmic reticulum (ER), mitochondria, and nucleus. Once synthesized, S1P is exported from cells through a S1P transporter (SPNS2 protein or ABC transporter) and subsequently binds to specific S1P receptors (S1PRs). These receptors trigger downstream signaling pathways in an autocrine or paracrine manner, known as inside-out signaling. Within the ER, S1P is subject to degradation by S1P lyase or recycled for the synthesis of ceramide (CER) and complex sphingolipids. Moreover, S1P can also be dephosphorylated by phosphatase 1 (SGPP1) and 2 (SGPP2), both located in the ER, to form Sph, which is also reused for CER synthesis. The activation of S1PR1-5 initiates G-protein mediated signaling pathways that govern various cellular processes, including migration, survival, morphogenesis, cell proliferation, immune trafficking, dendritic cell/macrophage activation, inflammation, and endothelial barrier regulation associated with the immune system, and barrier function. The figure was prepared using the Motifolio Illustration Toolkits (Motifolio Inc., Ellicott City, MD, USA). SMase, sphingomyelinase; CERase, ceramidase; CER synthase, ceramide synthase.

**Figure 2 ijms-24-12634-f002:**
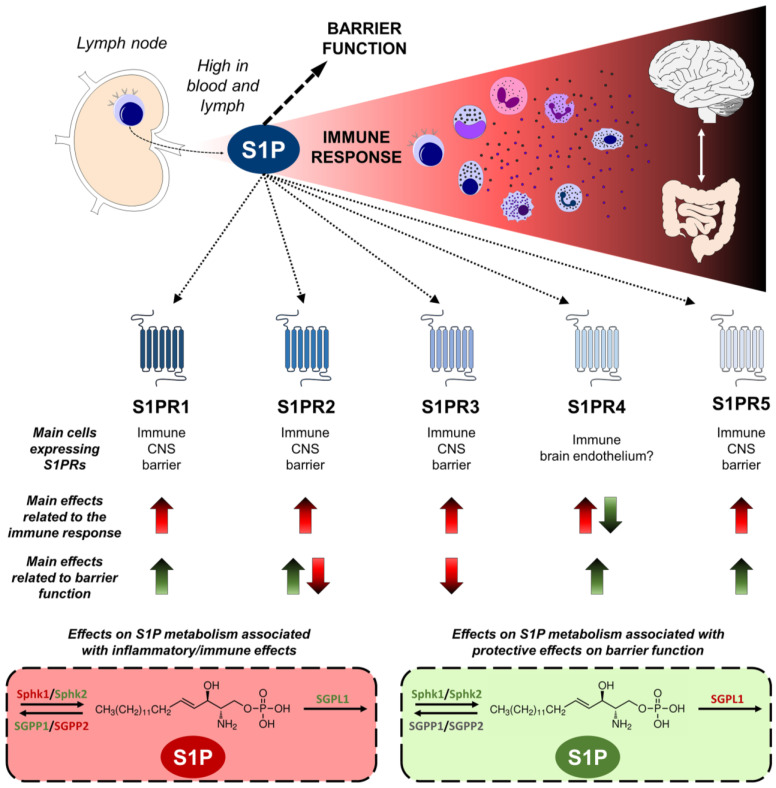
Sphingosine-1-phosphate (S1P) modulates the immune response and barrier function. S1P is a pivotal modulator of immune responses and barrier integrity. With elevated concentrations in the lymph and blood, S1P orchestrates immune cell trafficking from lymph nodes and contributes to inflammatory processes in several organs, including the central nervous system (CNS) and the gut, which are both relevant to psychiatric diseases. Five receptors (S1PR1-5) displaying distinct expression pattern in immune, CNS, and barrier-related cells, transduce the S1P signaling. Up arrows indicate an increase in the immune response or barrier function, and down arrows a decrease. Effects related to harmful situations are shown in red, and protective are in green. S1PR1, 2, 3, and 5 elicit inflammation and immune activation, while S1PR4 has shown ambivalent effects. The activity of S1PR1, 4, and 5 promotes a healthy barrier function, whereas S1PR3 exerts opposing effects, and S1PR2 yields controversial results. Similarly, S1P metabolism associated with inflammation includes the harmful activity of sphingosine kinase 1 (Sphk1) and sphingosine phosphatase 2 (SGPP2), and the suppression role of sphingosine kinase 2 (Sphk2), sphingosine phosphatase 1 (SGPP1), and S1P lyase 1 (SGPL1). Regarding protective effects on barriers, Sphk1 and 2 activation promote a healthier barrier, while SGPL1 is related to barrier dysfunction. The figure was prepared using the Motifolio Illustration Toolkits (Motifolio Inc., Ellicott City, MD, USA).

**Figure 3 ijms-24-12634-f003:**
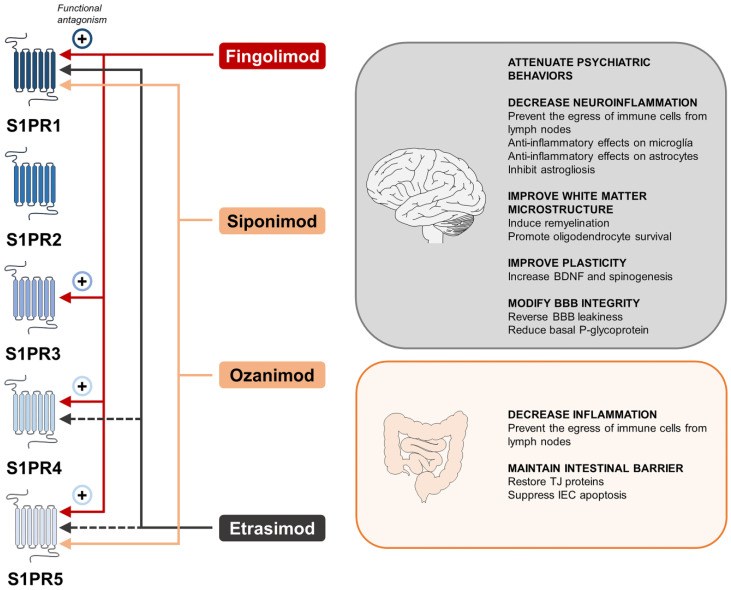
Effect of sphingosine-1-phosphate receptor (S1PR) modulators on S1PR isoforms and potential mechanisms in psychiatric and inflammatory bowel diseases (IBD). On the left, fingolimod binds S1PR1/3/4/5, siponimod and ozanimod are agonists of S1PR1/5, and etrasimod is a potent agonist of S1PR1 and a partial agonist of S1PR4/5. On the right, the potential biological processes modulated by these sphingosine-1-phosphate (S1P)-based drugs. The figure was prepared using the Motifolio Illustration Toolkits (Motifolio Inc., Ellicott City, MD, USA). BDNF, brain-derived neurotrophic factor; BBB, blood–brain barrier; TJ, tight junction; IEC, intestinal epithelial cell.

**Table 1 ijms-24-12634-t001:** Evidence of sphingosine-1-phosphate (S1P) in psychiatric diseases.

Authors and Year	Type of Study	Main Results
** *Major depressive disorder (MDD)* **		
Martín-Hernández et al., 2023 [229]	Preclinical (periodontitis + CMS)	S1P signaling modulation, decreased BBB-related proteins, and increased neuroinflammation
Shan et al., 2020 [228]	Preclinical (CMS)	S1PR2 was lower in the hippocampus, negatively correlated with symptom severity, and reduced TNF-α levels.
Corbett et al., 2019 [230]	Preclinical (chronic social defeat) Clinical (PTSD)	S1PR3 in the medial-PFC was elevated in resilient rats. S1PR3 was reduced in the blood of PTSD patients
Cheng et al., 2018 [231]	Preclinical (learned helplessness)	Fingolimod and TNF-α inhibition enhanced BBB integrity and ameliorate depressive behavior
Mike et al., 2018 [269]	Preclinical (neuropsychiatric lupus)	Fingolimod attenuated depressive-like behavior, neuroinflammation, and BBB permeability
Hunter et al., 2016 [270]	Clinical (MS)	Fingolimod improved depressive symptoms compared to other treatments in relapsing-remitting MS patients
Kornhuber et al., 2005 [271]	Clinical	Acid sphingomyelinase (ASM), which produces the S1P precursor ceramide, was increased in MDD patients
** *Neurodevelopmental disorders* **		
** *Schizophrenia (SZ)* **		
Li et al., 2023 [273]	Preclinical (cuprizone)	Fingolimod improved psychotic-behavior and decreased neuroinflammation.
Chand et al., 2022 [242]	Clinical	S1PR1 was higher in the dorsolateral-PFC of Type 2 SZ patients
Francis et al., 2021 [275]	Clinical	Fingolimod reduced circulating lymphocytes and improved white matter microstructure, but had no effects on symptoms in SZ patients
Esaki et al., 2020 [241]	Clinical	S1P was decreased in the *corpus callosum* of SZ patients
Patnaik et al., 2020 [286]	Preclinical (in vitro)	Fingolimod increased BDNF and promoted spinogenesis
Pépin et al., 2020 [249]	Preclinical (MPTP—PD model)	S1PR1 agonism ameliorated loss of dopaminergic neurons and motor deficits
Riganti et al., 2016 [247]	Preclinical	S1P induced synapsin I mobilization from synapses
Spampinato et al., 2015 [251]	Preclinical (in vitro)	Fingolimod reduced the immune-induced BBB damage
Miron et al., 2010 [281]	Preclinical (in vitro)	Fingolimod induced remyelination after demyelination
Miron et al., 2008 [280]	Preclinical (in vitro)	Fingolimod promoted extension and survival of oligodendrocytes
Coelho et al., 2007 [278]	Preclinical (in vitro)	Fingolimod exerted a cytoprotective effect and stimulation of remyelination on oligodendrocytes
Jung et al., 2007 [279]	Preclinical (in vitro)	Fingolimod improved the survival of rat oligodendrocytes
** *Autism spectrum disorders (ASD)* **		
Almandil et al., 2023 [258]	Clinical	Single nucleotide polymorphisms (SNPs) in the Sphk1 gene were associated with ASD in females
Li et al., 2022 [288]	Preclinical (BTBR mouse strain)	SphK blocker SKI-II recovered behavioral and molecular impairments
Naegelin et al., 2021 [285]	Clinical (Rett syndrome)	Fingolimod did not show behavioral and biochemical effects in children with RS
Kurochkin et al., 2019 [257]	Clinical	S1P levels were unaltered in the PFC of ASD patients
Gurevich et al., 2018 [277]	Clinical (MS)	Fingolimod improved white matter microstructure in MS patients with impaired pyramidal function
Wu et al., 2018 [259]	Preclinical (valproic acid)	S1P was decreased in the hippocampus and serum and there were memory impairments
Wang et al., 2016 [256]	Clinical	S1P serum levels were reduced in ASD patients
Wang et al., 2013 [276]	Preclinical (EAE—MS model)	Fingolimod ameliorated white matter microstructure
** *Attention deficit hyperactivity disorder (ADHD)* **		
Brunkhorst-Kanaan et al., 2021 [265]	Clinical	S1P plasma levels were increased in ADHD adults
Henriquez-Henriquez et al., 2020 [263]	Clinical	Alterations in ceramide synthesis genes, a precursor of S1P in ADHD patients
Henriquez-Henriquez et al., 2015 [264]	Clinical	Lower levels of sphingomyelins in ADHD children

Abbreviations: chronic mild stress, (CMS); blood–brain barrier, (BBB); sphingosine-1-phoshpate receptor, (S1PR); tumor factor necrosis alpha, (TNF-α); posttraumatic stress disorder, (PTSD); prefrontal cortex, (PFC); multiple sclerosis, (MS); brain-derived neurotrophic factor, (BDNF); 1-methyl-4-phenyl-1,2,3,6-tetrahydropyridine, (MPTP); sphingosine kinase, (Sphk); experimental autoimmune encephalitis, (EAE).

**Table 2 ijms-24-12634-t002:** Evidence of sphingosine-1-phosphate (S1P) in inflammatory bowel diseases (IBD).

Authors and Year	Type of Study	Main Results
** *Inflammatory bowel diseases (IBD)* **		
Sandborn et al., 2023 [293]	Clinical (UC)	Etrasimod was effective and safe as an induction and maintenance therapy in moderate to severe UC patients.
Martín-Hernández et al., 2022 [88]	Preclinical (sub-chronic stress)	Stress increased S1P, dysregulated immune responses, and enhanced intestinal permeability. Sphk2^−/−^ mice presented a cytokine-expression profile towards a boosted Th17 response, lower expression of claudins, and structural abnormalities in the colon.
Sandborn et al., 2021 [291]	Clinical (UC)	Ozanimod exerted long-term (2 years) benefits and was safe (4 years) in moderate to severe UC patients
Sandborn et al., 2021 [292]	Clinical (UC)	Ozanimod was effective and safe as an induction and maintenance therapy in moderate to severe UC patients.
Kiyomi et al., 2020 [294]	Clinical	Etrasimod reduced circulating adaptive but not innate immune cells in healthy individuals.
Chen et al., 2018 [218]	Preclinical (DSS-induced colitis)	Intestinal barrier damage was higher in S1PR2^−/−^. S1P/S1PR2 axis mediated CD4+T cell activation via ERK and MHC-II in IECs
Karuppuchamy et al. 2017 [221]	Preclinical (DSS-induced colitis)	Chronic inflammation modulated S1PR1 expression and tissue S1P levels.
Dong et al., 2015 [219]	Preclinical (IL-10^−/−^ mice)	S1PR1 agonism improved barrier function by restoring TJ proteins and suppressing IEC apoptosis
Montrose et al., 2013 [220]	Preclinical (DSS-induced colitis) Clinical (UC)	S1PR1 deletion enhanced vascular permeability and bleeding in mice. Patients with active UC showed overexpression of S1PR1 and increased vascular density in inflamed the colon mucosa.
Eisenhoffer et al., 2012 [217]	Preclinical (in vitro)	S1PR2 receptor and ROCK controlled IEC extrusion, critical for a healthy intestinal barrier function
Greenspon et al., 2011 [216]	Preclinical (in vitro)	S1P increased E-cadherin levels, enhancing barrier function in cultured IECs
Sandbornd et al., 2016 [290]	Clinical (UC)	Ozanimod exerted a slightly higher rate of clinical response and remission (preliminary trial) in moderate to severe UC patients.

Abbreviations: ulcerative colitis (UC), sphingosine kinase (Sphk), T helper lymphocyte (Th17), dextran sulfate sodium (DSS); sphingosine-1-phoshpate receptor (S1PR), extracellular signal-regulated kinase (ERK), major histocompatibility complex (MHC), intestinal epithelial cell (IEC), interleukin-10 (IL-10), tight junction (TJ), Rho kinase (ROCK).

## Data Availability

Not applicable.

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
