# Peer review of "Immune System and Brain/Intestinal Barrier Functions in Psychiatric Diseases: Is Sphingosine-1-Phosphate at the Helm?"

_ijms, 2023, doi:10.3390/ijms241612634_

Round 1

Reviewer 1 Report

This manuscript by Martín-Hernández et al as a review article provides a detailed narrative on the role of S1P in psychiatric diseases. The theoretical basis centers on the documented functions of S1P on the regulation of immune system and brain and intestinal barrier integrity. While the review of sphingolipid metabolism and its derivatives, such as S1P, in different disease contexts has been frequently seen in the literature, the focus on its functional relevance and significance in psychiatric diseases and various phenotypes is timely, informative and significant. The authors are commended for providing a comprehensive coverage of the current knowledge as it pertains to the S1P’s function in immune regulation and in psychiatric diseases.

It is evident from the narrative, S1P’s function is pleiotropic and involves a complex network of regulation, but the current experimental data and, to some extent, those from human studies, do indicate the potentially critical role of S1P in psychiatric diseases. The authors’ thought-provoking argument that its primary functional impact may involve the control of brain and intestinal barrier homeostasis, in which the “leaky gut” phenomenon may be directly relevant, is interesting and important. Overall, this review is well written and pertinent sections are nicely laid out. Two schematic diagrams illustrating S1P signaling, metabolism and functions are clear and to the point. Also, the cited reference is up-to-date. Nevertheless, the following two issues may require consideration for further elaboration and clarification:

1.      It would be ideal to provide a summary table listing those major controversial findings and include a paragraph or two stating the likely reasons for those inconsistent experimental findings. Also, the review content can be further strengthened by expanding the expert opinions on future direction in this area of research and development.

2.      The study of S1P’s function is challenging due, in part, to the complexity in its regulatory network involving different cell types, receptors, intracellular and extracellular activities. While the authors have provided experimental accounts supporting the role of S1P’s function, particularly regarding its function in regulating immunity and barrier integrity, it is, at present, still uncertain as to whether S1P and its function is required and/or sufficient to explain the pathological phenotypes, considering also the fact that certain degree of heterogeneity in disease is anticipated. It would be further strengthened by emphasizing and elaborating further some of the most significant findings in the literature documenting the relevant role of S1P in psychiatric diseases. Also, a question “…is sphingosine-1-phosphate at the helm?” is stated in the title, but following the read through the text, it is unclear as to whether S1P is, in fact, "at the helm" or it is simply involved in the disease process? Modifying the title or stating clearly the answer to the question, perhaps, at the end of the text as a summary is needed.  

Author Response

We sincerely appreciate the valuable comments from the reviewer, as he/she has raised critical points. Given that this is a narrative review, we have carefully considered his/her suggestion, and we believe that incorporating a summary table with the controversial findings mentioned in our manuscript may not sufficiently represent the complexity of the subject matter and could potentially be misleading for the readers. However, the question is utterly relevant and needs to be addressed. Thus, we have elaborated on the controversial studies in S1P research as suggested by the reviewer trying to shed some light on the feasible explanations behind them. Additionally, we have delved more into the future perspectives of S1P research in psychiatric diseases focusing on the most critical points to finally unravel its pharmacological possibilities. Finally, a part is dedicated to answer the question formulated in the title. Consequently, section 7 has been almost entirely rewritten to include all this information:

This review has examined the current evidence demonstrating the ability of S1P to modulate the immune system and barrier functions (BBB and intestinal), processes that may underlie relevant pathophysiological features to psychiatric diseases. S1P signaling has emerged as a promising pharmacological target, particularly due to the proven efficacy of approved drugs in treating CNS and intestinal pathologies. So, is S1P at the helm of immune and barrier mechanisms in psychiatric diseases? It is not yet possible to give a conclusive response without first elucidating some critical points through prospective studies.

Addressing these questions poses several difficulties. Firstly, the exact molecular mechanisms underlying S1P signaling are unknown despite numerous research efforts. Although S1P is generally recognized to increase immune system activity and maintain barrier function, controversial data hinder a conclusive understanding of the role of some S1P elements due to complex signaling involving specific cell types and intricate intracellular and extracellular pathways. For instance, S1PR1 stimulation in the context of psychiatric diseases can be seen as deleterious for increasing inflammation but beneficial for promoting barrier integrity. More mechanistic studies are mandatory to understand the molecular functioning of all the S1P pathway elements and explain the big picture.

Secondly, pre-clinical and clinical evidence primarily relies on non-specific S1PR modulators, making it challenging to attribute their effects to specific isoforms or cell types. New pharmacological tools specific to each S1P element could help to unravel the complex network signaling of S1P. Despite this limitation, the significant studies gathered in this review show promising actions of available S1P-based drugs in psychiatric diseases, demonstrating improvements in behavioral and pathophysiological features. While S1P clinical evidence is still limited, it reinforces the idea of S1P playing a role in these illnesses.

The final main difficulty is inherent to the nature of MDD, SZ, or ASD as multifactorial and heterogeneous diseases with elusive pathophysiology. Neuroinflammation and barrier alterations are transversal to all these pathologies, but they may merely be coincidental phenomena. If this were the case, the possibilities of S1P as a pharmacological alternative would decrease unless it modulates other essential mechanisms beyond immune and barrier functions. Moreover, relevant differences exist among psychiatric illnesses and even their subtypes. Therefore, future research should carefully analyze which pathologies or subtypes would benefit from controlling inflammation and barrier integrity to achieve clinical improvement.

In light of all the exposed reflections, the question in the title remains open but highly relevant. While a definite role of S1P in psychiatric diseases cannot be ascribed yet, emerging evidence, particularly in the last years, augurs a necessary intensification of research in this field. Targeting S1P has the potential to modulate immune and barrier alterations associated with these diseases and may have implications in their possible origin through the leaky gut phenomenon. As our understanding of the intricate signaling pathways and molecular mechanisms involving S1P continues to expand, so does the potential to target this lipid mediator for more effective and specific treatments in various psychiatric conditions. Innovative strategies, such as selective modulators of S1P receptors, combination therapies, and personalized medicine approaches, may usher in a new era of precision psychiatry. While challenges lie ahead, the ongoing pursuit of S1P-related research holds immense promise in alleviating the burden of psychiatric diseases and improving the lives of millions worldwide.

Reviewer 2 Report

David et al. summarized current research finding about the role of S1P and S1PRs in psychiatric diseases and inflammatory bowel diseases. This is a good review paper with novelty and significance. I have some suggestions to the authors.

1. It is better to include a new figure to describe the effects of S1PR modulators and potential mechanisms in treating psychiatric diseases and inflammatory bowel diseases.

2. The authors should described clearly S1PR modulators (fingolimod, ozanimod). Which S1PRs are affected by them, and if the receptors are activated or suppressed by these S1PR modulators.

Author Response

We thank the reviewer for his/her insightful observations. We acknowledge that the previous version of the manuscript lacked information about the S1PR modulators. Thus, we decided to clarify their direct effects on the S1PRs and the potential actions relevant to psychiatric diseases and IBD by including a new paragraph and a Figure (Figure 3) at the beginning of section 6:

Several compounds differentially target the isoforms of the S1PRs, having all in common the functional antagonism of S1PR1 in immune cells through its internalization. Among these drugs, fingolimod, siponimod, ozanimod, and etrasimod have garnered attention in this review due to their actions. Fingolimod was the pioneering development, displaying a broader receptor affinity with effects on S1PR1, 3, 4, and 5. Subsequent compound designs have emphasized enhancing specificity while minimizing adverse side effects. Siponimod selectively targets S1PR1 and S1PR5, avoiding the activation of S1PR3 {Gergely, 2012 #4146}, and ozanimod was developed later as a potent agonist of S1PR1 and 5, with residual activity over the S1PR2,3, and 4 {Scott, 2016 #4145}. On the other hand, etrasimod acts as a full agonist of S1PR1 and a partial agonist of S1PR4 and 5 {Buzard, 2014 #4147}. Figure 3 illustrates the activity of S1PR modulators on the different isoforms of the S1PRs summarizing the evidence gathered in this review relevant to psychiatric diseases and IBD.

Figure 3

Figure 3: Effect of sphingosine-1-phosphate receptor (S1PR) modulators on S1PR isoforms and potential mechanisms in psychiatric and inflammatory bowel diseases (IBD). On the left, fingolimod binds S1PR1/3/4/5, siponimod and ozani-mod are agonists of S1PR1/5, and etrasimod is a potent agonist of S1PR1 and a partial agonist of S1PR4/5. On the right, the potential biological processes modulated by these sphingosine-1-phosphate (S1P)-based drugs. BDNF, brain-derived neurotrophic factor; BBB, blood-brain barrier; TJ, tight-junction; IEC, intestinal epithelial cell.
